# Internal Jugular Vein Thrombosis: Etiology, Symptomatology, Diagnosis and Current Treatment

**DOI:** 10.3390/diagnostics11020378

**Published:** 2021-02-23

**Authors:** Alba Scerrati, Erica Menegatti, Matilde Zamboni, Anna Maria Malagoni, Mirko Tessari, Roberto Galeotti, Paolo Zamboni

**Affiliations:** 1Unit of Neurosurgery, University Hospital of Ferrara, 44121 Ferrara, Italy; scrlba@unife.it; 2Vascular Diseases Center, University Hospital of Ferrara, 44121 Ferrara, Italy; mngrce@unife.it (E.M.); anna.maria.malagoni@unife.it (A.M.M.); mirko.tessari@gmail.com (M.T.); zambo@unife.it (P.Z.); 3Unit of Vascular and Endovascular Surgery, Hospital of Mestre, 30100 Venice, Italy; 4Unit of Vascular and Interventional Radiology, University Hospital of Ferrara, 44121 Ferrara, Italy; gtt@unife.it

**Keywords:** internal jugular vein, venous thromboembolism, deep veins thrombosis, brain circulation, cerebral venous drainage

## Abstract

(1) Background: internal jugular vein thrombosis (IJVthr) is a potentially life-threating disease but no comprehensive reviews on etiology, symptomatology, diagnosis and current treatment guidelines are yet available; (2) Methods: we prospectively developed a protocol that defined objectives, search strategy for study identification, criteria for study selection, data extraction, study outcomes, and statistical methodology, according to the PRISMA standard. We performed a computerized search of English-language publications listed in the various electronic databases. We also retrieved relevant reports from other sources, especially by the means of hand search in the Glauco Bassi Library of the University of Ferrara; (3) Results: using the predefined search strategy, we retrieved and screened 1490 titles. Data from randomized control trials were few and limited to the central vein catheterization and to the IJVthr anticoagulation treatment. Systematic reviews were found just for Lemierre syndrome, the risk of pulmonary embolism, and the IJVthr following catheterization. The majority of the information required in our pre-defined objectives comes from perspectives observational studies and case reports. The methodological quality of the included studies was from moderate to good. After title and abstract evaluation, 1251 papers were excluded, leaving 239 manuscripts available. Finally, just 123 studies were eligible for inclusion. We found out the description of 30 different signs, symptoms, and blood biomarkers related to this condition, as well as 24 different reported causes of IJVthr. (4) Conclusions: IJVthr is often an underestimated clinical problem despite being one of the major sources of pulmonary embolism as well as a potential cause of stroke in the case of the upward propagation of the thrombus. More common symptoms are neck pain and headache, whereas swelling, erythema and the palpable cord sign beneath the sternocleidomastoid muscle, frequently associated with fever, are the most reported clinical signs. An ultrasound of the neck, even limited to the simple and rapid assessment of the compression maneuver, is a quick, economic, cost-effective, noninvasive tool. High quality studies are currently lacking.

## 1. Introduction

The internal jugular vein (IJV) originates at the jugular foramen, runs along the lateral neck, medially to the sternocleidomastoid muscle from the carotid triangle, and ends at the brachiocephalic vein. The IJV is one of the four components of the vascular sheath of the neck, together with the common and internal carotid arteries, the vagus nerve, and the deep cervical lymph nodes. IJV thrombosis (IJVthr) is a potentially life-threating disease, due to the development of an intraluminal thrombus. The thrombosis can occur at any level, from the jugular foramen to the junction between the IJV and the subclavian vein, constituting the brachiocephalic vein [1,2,3]. Serious life-threatening complications have been described following IJVthr, such as pulmonary embolism (PE), lower airway swelling, superior sagittal sinus thrombosis, intracranial hypertension, cerebral oedema, septic emboli, chylothorax, and superior vena cava syndrome [3,4,5,6,7].

## 2. Materials and Methods

### 2.1. Study Design

We prospectively developed a protocol that defined the objectives of this review, a search strategy for study identification, criteria for study selection, data extraction, study outcomes, and statistical methodology. We followed the PRISMA criteria for systematic reviews [8]. 

The present systematic review was addressed to define clinical signs, symptoms, diagnostic tools, complications and treatment of IJVthr in clinical practice. 

### 2.2. Data Sources and Searches

We performed a computerized search of English-language publications listed in the electronic databases (PubMed, EMBASE, Science Citation Index, and Google Scholar) up to July 2020 using the following text and key words in combination, both as MeSH terms and text words: [“internal jugular vein thrombosis”] and [“autoptic studies”, “CT scan”, “epidemiology”, “ethiology”, “complications”, “imaging”, “intracranial veins”, “MRI”, “pulmonary embolism”, “sympthomatology”, “therapy”, “treatment”, “ultrasound”, “COVID19”]; [“internal jugular vein obstruction”] and [“CT scan”, “ethiology”, “complications”, “imaging”, “intracranial veins”, “MRI”, “sympthomatology”, “thrombosis”, “ultrasound”]; [“superior vena cava syndrome”]; [“central venous catheter”] and [“complications”]; and [“randomized controlled trial”, “cohort studies”, “prospective studies”, “clinical trial”, “case report”]. Full texts and their references lists (forward search) were evaluated for inclusion. Finally, we retrieved potentially relevant reports from other sources, especially by the means of hand search in the Glauco Bassi Library at the University of Ferrara [9].

### 2.3. Study Selection

One investigator (PZ) assessed articles for eligibility. If the title and the abstract were judged to be potentially eligible, he assigned the full article to two reviewers (EM, AS) in order to extract the required data. 

The inclusion criteria were as follows: (1)studies reporting patients with an objective diagnosis of IJVthr;(2)studies correctly addressing the question of the search reported in the abstract;(3)studies reporting the related clinical outcome.

Studies satisfying all the inclusion criteria were selected.

### 2.4. Data Extraction and Quality Assessment

EM and AS independently extracted data using a standardized form. The data were extracted for study design, study quality, number, sex and age of enrolled patients, duration of follow-up. and outcome measures.

We assessed the methodological quality of included studies using criteria adapted from guidelines for the evaluation of articles on therapy and prognosis [10].

Quality assessment of the included studies was performed using the Newcastle–Ottawa Scale (NOS) for assessing the quality of non-randomized studies in meta-analyses (http://www.ohri.ca/programs/clinical_epidemiology/oxford.asp accessed on 20 February 2021).

The NOS contains eight items, categorized into three dimensions including selection, comparability, and—depending on the study type—outcome (cohort studies) or exposure (case-control studies). For each item, a series of response options is provided. A star system is used to allow a semi-quantitative assessment of study quality, such that the highest quality studies are awarded a maximum of one star for each item with the exception of the item related to comparability that allows the assignment of two stars. The NOS ranges between zero up to nine stars.

## 3. Results

### 3.1. Study Identification and Selection

Using the predefined search strategy, 1490 titles were retrieved and scanned. Data from randomized control trials were few and limited to the central vein catheterization route and material and to the IJVthr anticoagulation treatment. No systematic reviews were available, except for Lemierre syndrome and IJVthr, and jugular thrombosis following catheterization. The majority of information required in our pre-defined objectives comes from perspectives observational studies and from case reports. After title and abstract reading, 1251 papers were excluded, leaving 239 manuscripts for full reading. Finally, 123 studies were eligible for inclusion in the present review, as shown in Figure 1.

### 3.2. Study Quality

The scores of the included papers, retrieved using the NOS, ranged between 5 and 7 stars (33 studies 5 stars, 58 studies 6 stars, 32 studies 7 stars). According to this scale, we can consider the general quality of the papers moderate/good. Papers reporting symptoms, clinical signs, and rare causes of IJVthr are case series of lower quality, whereas articles related to central venous catheter and treatment were better scored.

### 3.3. Clinical Presentation 

Our results show the most common symptoms of IJVthr include neck pain [2,3,11,12,13,14,15,16] and headache (see Table 1) [12,16,17,18,19,20,21,22]. 

There were several common clinical signs (Table 1). The most frequent were neck swelling and erythema [2,3,5,6,11,12,13,14,15,17,23,24,25,26,27] and the palpable cord beneath the sternocleidomastoid muscle [2,3,6,28], frequently associated with fever [3,14,18,28]. Oedematous swelling of the face/scalp [17,18,26,29,30], papilledema [12,14,17,22] and intracranial hypertension [9,12,14,22,28] were also reported in several papers. These kinds of symptoms could be ascribed to a possible impairment of the cerebrospinal fluid dynamics, which is strictly connected to the cerebral venous outflow [31].

Laboratory results (Table 1) showed the D-dimer raising [19,25,28] and leukocytosis [3,5,14,25,26,28] were the most common abnormalities. Quite recently, a number of papers focused on symptoms related to IJV obstruction, especially from bone and muscular compression but even from kinking of the carotid arteries. IJV obstruction from bone nutcracker represents a risk factor for the development of IJVthr [32]. However, other non-specific symptoms of IJVthr/obstruction, with or without thrombosis, are the following: headache, numbness, eye discomfort and blurred vision, sleep disorders, tinnitus, hearing loss, dizziness, and demyelination. Finally, several papers showed that IJV proximal obstruction is a risk factor for non-aneurysmal peri-mesencephalic hemorrhage and stroke [32,33,34,35].

### 3.4. Post-Mortem Studies

IJVthr is clinically considered a rare event if compared to data coming from post-mortem studies, especially in hospitalized patients. Several clinical-pathological studies have been published since the early 1960s [38,39,40]. 

The most recent one on 5039 autoptic cases collected in several German hospitals, deep vein thrombosis (DVT) was found out in 1725 cases (34.9%). In 404 cases it was found in the superior vena cava system, in 730 cases in the inferior vena cava system and in 248 cases in combined localization [41]. Pulmonary dissection revealed embolism in 1500 cases (29.8%), of which it was fatal in 628 cases (41.9%). In 12.6% the thrombosis was present in the upper system while in the lower extremity was found 5 times (59.4%) more common. Interestingly, after the routinely introduction of the central veins’ catheterization, the location of DVT was found in the right IJV in 30.9%, in the left IJV in 15.7%, in the left femoral vein in 32.7%, in the right femoral vein in 30% and in the deep calf veins in 15.1% [41].

By comparing autopsy studies of different times [38,39,40], the change in the principal localizations of venous thrombosis is the most striking feature. In Berlin, at the beginning of 20th century, thrombi in jugular veins were reported with a very low frequency (1.7%) [41]. This constitutes an evidence for a causal role of central veins catheterization for the observed shift to the superior vena cava system. Thanks to its good accessibility, the right IJV is a widely favorite site for the insertion of central venous catheters, explaining why such location challenges the primate to the femoral vein [42,43,44,45]. One of the limit of this paper is that it is very difficult to differentiate if the upper vein thrombosis includes the other etiology besides IJVthr.

### 3.5. Etiology 

IJVthr is often a complication of head and neck, local skin, and throat infections [23,24,46,47,48], surgery [49,50,51], trauma [17], local or distant malignancy [3,11,25,51,52], central venous catheter placement [43,44,45], polycythemia vera [37,53,54,55], intravenous drug abuse [56,57], neck massage [12], ovarian hyperstimulation syndrome [58,59,60,61], hypercoagulable state secondary to factor V Leiden, protein C, protein S, anti-phospholipid syndrome, anti-thrombin III deficiency [13,14,36,62,63,64,65], or it can be a primary process [1,3,4,5]. 

The main cause of IJVthr are shown in Table 2.

As above reported, central veins catheterization contributes to the increasing prevalence and incidence of IJVthr in hospitalized patients. On the other hand, both IJV and femoral vein are suitable routes for catheters when a subclavian approach is not feasible. A randomized control trial clearly indicates that femoral vein catheterization is associated with a 25% higher frequency of lower extremity deep vein thrombosis compared to similar patients receiving subclavian or IJV catheters [78]. 

Catheter infection resulted significantly correlated to thrombosis and could constitute a primary risk factor. It is a clinical parameter with a high sensitivity (86%) and an equal specificity (97%) of erythema, tenderness, warmth, or swelling in predicting the development of IJVthr [45]. According to a randomized control trial, both femoral and IJV accesses lead to similar risks of catheter infection. However, the findings of the same study recommend IJV for female, non-chlorhexidine-impregnated dressings users, and when catheters are left in place more than 4 days [79].

IJVthr related to catheterization is particularly frequent in the end-stage renal disease patients. Since renal transplantation is not suitable for around 60% of such patients, hemodialysis is the next therapeutic option. However, it requires good venous access sites with at blood flow of at least 350 mL/min. In those patients, vascular accesses are long-lasting, so an arteriovenous fistula (AVF) is the most appropriate technique. When an AVF is not technically feasible or when patients require dialysis during the maturation of the AVF, a hemodialysis catheter in the central veins represents the most widely used option [29,80,81,82,83].

Patient-related factors have found to increase the risk of catheter related IJVthr. For instance, older age, high body mass index, cancer, and patients in ICU have been associated with an increased risk, whereas sex and ethnicity have not. The presence of inherited thrombophilia and/or a personal history of venous thromboembolism of course increase the risk [84].

Oncological patients are the second high-risk category for IJVthr with or without central veins catheterization [3,11,25,51,52]. Data from randomized clinical trials do not support a favorite route in terms of IJVthr development [72,73,74], thus the complication seems to be more related to certain type of malignancy rather than to the catheterization site. Particularly recurrent in the literature is gastric cancer with Trousseau metastasis in the left neck lymphatic system, which indicates neoplastic cell delivery in the IJV through the thoracic duct, ending in the left IJV [11,25,85]. Distant malignancy with paraneoplastic IJVthr was found in 54% of patients admitted to a large ENT department, whereas in 68% of cases the tumor was located in the head and neck region. Finally, approximately one third of the patients with distant tumor entities had thrombosis of the IJV as the first symptom of the disease [15,28]. In a retrospective review of almost 2000 cases of DVT, the process was found out to be located in the IJV in 1.5% [2].

### 3.6. Other Less Common Causes

IJVthr in medical literature has been associated to a variety of different conditions, generally described in anecdotal case reports and in safety endovascular studies: effort thrombosis of the axillary/subclavian venous trunk extended to the IJV [2]; the combination of congenital IJV hypoplasia with inherited coagulation deficit [50]; in microgravity condition, during space flight on the International Space Station, for the combination of dehydration status with the lack of gravitational gradient [76,77]; rarely following balloon venoplasty but more frequently after stenting of the IJV for chronic cerebrospinal venous insufficiency (CCSVI) treatment [66,67,86,87]; an elongated styloid process [24,88,89] and associated to Behçet disease [26].

In rare cases, IJV surgical manipulation during skull base approaches for the mastoid or cervical region should be considered [90,91,92,93,94].

Finally, truncular IJV malformation typical of CCSVI can be complicated by thrombosis (Figure 2) [29,61,95]. However, in a wide retrospective case revision, in half of the cases IJVthr was considered idiopathic [2].

### 3.7. Pulmonary Embolism and Other Serious Complications 

Patients with IJVthr often receive anticoagulation in consequence of the risk of further potentially life-threatening PE [96].

The reported rate of PE occurring in case of IJVthr goes from 1.6% [97] to 22.7% [98], although this percentage is taken from small and not recent retrospective studies. PE ranged from 0.5% in case of isolated IJVthr to 2.4% when combined with subclavian/axillary vein thrombosis. The morbidity and mortality of IJVthr seem quite similar to that of the upper extremity deep vein thrombosis; accordingly, consideration should be given to treat these two entities in a similar way [3,6]. 33% of hospitalized patients evaluated by the means of ultrasound in the superior vena cava system were found to have IJVthr, eventually coupled with other venous location in the chest and/or in the upper arm. Again, the presence of a central venous catheter was the only factor found to be a significant risk factor. Among these, 8% had a PE documented by computed tomographic angiography/pulmonary arteriography, but never fatal. According to a recent systematic review, concomitant thrombosis of the axillary vein or the subclavian vein introduces a second possible source of embolization. Therefore, determining the theoretical risk of PE from an isolated IJVthr is confounded by the presence of concomitant thrombosis in other veins and, currently, there is little proof of the propagation of the thrombus to cause a clinically overt PE in case of isolated IJVthr [99]. 

However, data from autopsy studies seem to tell us that IJVthr can be misdiagnosed because of often presenting with mild-moderate symptomatology [41,43].

Furthermore, our systematic review found out other severe complications of IJVthr, potentially life threating, described in case reports: chylotorax [16,69,75,94,100], airways swelling, Lemierre syndrome and septic emboli [101,102,103], and superior vena cava syndrome [18,30,70].

### 3.8. Combination of Internal Jugular Vein and Cerebral Veins Thrombosis

Thrombosis of the cerebral veins (CVT) is an uncommon form of stroke, usually affecting young individuals [104].

Multiple factors have been associated with CVT: thrombophilia, inflammatory bowel disease, pregnancy, dehydration, infection, oral contraceptives, substance abuse, and unpredictable events (e.g., head trauma).

Several reports described CVT due to the propagation of IJVthr into the intracranial system [19,20,21,88,105,106].

Recently, cerebral venous sinus thrombosis as a presentation of COVID-19 has been described [107]. Coronavirus disease 19 (COVID-19) is primarily a disease with respiratory manifestations, but there are increasing reports of cardiovascular and thromboembolic complications [108,109,110,111,112].

Clinical findings are related to the increased intracranial pressure attributable to impaired venous drainage and to focal brain injury from venous ischemia/infarction or hemorrhage [113].

Headache, generally indicative of an increase of the intracranial pressure, is the most common symptom and is typically described as diffuse and often progresses in severity over days to weeks. Papilledema or diplopia (caused by sixth nerve palsy) can occur [22].

Neurological signs and deficits occur in case of venous ischemia or hemorrhage. The most common are hemiparesis and aphasia. Scalp edema and dilated scalp veins may be seen on examination [114].

Moreover, first, focal, or generalized seizures are frequent, occurring in ≈40% of patients. Finally, patients with CVT often present with slowly progressive symptoms. Delays in the diagnosis of CVT are common and significant. 

### 3.9. Ultrasound in IJVthr Screening and Follow up 

The examination of the extracranial cerebral venous system by means of color Doppler ultrasound (US) is aimed to determine either the insufficiency of such a venous system [105], or the obstruction secondary to the presence of thrombosis, adding an accurate compression ultrasound (CUS) maneuver [106]. The IJV investigation starts with the patient in supine position keeping head facing forward, avoiding flexion, hyperextension, and rotation of the neck (these postures may compress the veins and influence the measurements). Moreover, the operator should use the lightest possible pressure on the neck in order to avoid IJV flattening [115].

It is mandatory to evaluate both in the transverse and longitudinal plane the IJV morphology and hemodynamic at three different levels:

J1-proximal portion of the jugular vein and the subclavian/innominate vein

J2-level of the thyroid gland

J3-level of the carotid bifurcation.

Regarding the IJV patency assessment performed by CUS using B-mode imaging, it must be done all along the vessel from J1 to J3 in transverse plan. Power Doppler and/or B-Flow are useful tools that can provide information about the flow surrounding the thrombus, even the slowest one, as shown in Figure 2 [116,117,118].

We found articles evaluating the accuracy of CUS in the diagnosis of IJVthr although the assessment includes cumulatively venous segments of the upper extremities together with the jugulars [119,120].

Using venography as the reference method, the sensitivity and specificity of both CUS (96% and 93.5%, respectively) and color flow Doppler imaging (100% and 93%, respectively) were highly reliable [121].

Finally, in catheter related IJVthr, ultrasound monitoring is highly recommended [84].

### 3.10. Future Perspectives in Non-Invasive IJVthr Diagnostic

Flow in large vessels depends on the cardiac and respiratory cycles being not constant over time, particularly for a pulsatile vein such as the IJV, where cardiac pump transmits a sequence of well described waves along the entire vessel. This phenomenon is the ‘so called’ jugular venous pulse (JVP) and it is one of the main parameters of cardiac function. Indeed, it is also the reference physiological signal used to detect right atrial and central venous pressure (CVP) abnormalities in cardio-vascular diseases (CVDs) diagnosis [111]. Recently a novel technique has been described to evaluate the JVP by means of US, consisting of an accurate measurement of IJV CSA variation during cardiac cycle [115,122,123]. The protocol consisted in a IJV B-mode ultrasound videoclip recording, synchronized with 3-lead ECG, setting the frame rate at 30 frames/min in order to have the better time resolution. The assessment of the IJV CSA was performed using transverse plane at C5-C6 level [115,122].

The curve produced by the elaboration of CSA variation over the individual cardiac cycle, in case of physiological venous drainage, exactly corresponds to the JVP. Moreover, JVP trace permits a reliable and non-invasive estimation of central venous pressure. Using the lagging autocorrelation r-values as predictors, mean-CVP was predicted from ultrasound JVP with a mean-absolute-error of 1.45 cmH_2_O. Considering that JVP reflects right atrial pressure measurement, being directly connected to IJV without interruption [124] this curve could predict the presence of obstructions within the atrium-jugular segment. The presence of thrombus inside the IJV, subclavian vein, or superior vena cava would prevent the correct pressure wave propagation from the heart to the jugular, leading a change in the US-JVP curve. Further studies will be needed to demonstrate the changing in ultrasound JVP trace in course of IJVthr as shown in Figure 2.

Another non-invasive and cost-effective tool for the investigation of cerebral venous drainage is constitute by cervical strain-gauge plethysmography. This technique was demonstrated to differentiate healthy subjects (HC) and patients affected by CCSVI in terms of empty gradient and empty time, highlighting the faster outflow velocity and time in HC respect to the control [125]. Cervical strain-gauge plethysmography was also used to measure the cerebral venous return in microgravity, monitoring the essential compensatory mechanisms activated in subjects living in this special environment. Comparing pre, post and in-flight data confirmed a redistribution of venous blood volume during spaceflights mission, showing differences in the amplitude of cardiac pressure wave propagation measured at neck veins level [126]. 

Moreover, a recent study described neck photoplethysmography as an alternative for extracting the JVP from the anterior jugular veins, indicating it as a significant tool for the future of cardiovascular diagnosis [127]. 

All these studies pave the way for future application of strain-gauge plethysmography as a possible first level screening device in case of IJVthr, considering the cut-off values of the patients affected by extracranial venous obstruction and the possibility to trace the characteristic JVP curve with this novel technique [125,126].

### 3.11. Second Level Diagnosis 

Head CT without contrast is often normal but may demonstrate findings that suggest cerebral veins thrombosis (CVT) like hyperdensity of a cortical vein or dural sinus (approximately one third of patients) [128]. An ischemic infarction that usually crosses the arterial boundaries (particularly with a hemorrhagic component) or proximal to a venous sinus are suggestive of CVT [120]. Subarachnoid hemorrhage and ICH are infrequent [129]. Contrast-enhanced head CT scan may show enhancement of the dural lining of the sinus with a filling defect within the vein or sinus.

MRI has a sensitivity and specificity of nearly 100% in assessing the patency of the superior vena cava and the internal and external jugular veins. However, the sensitivity drops to 83% in the shoulder area [130].

The MR signal intensity of venous thrombus varies according to the time of imaging from the onset of thrombus formation [128]. In the acute stage of thrombus formation (0–5 days), the signal is predominantly isointense on T1-weighted images and hypointense on T2- weighted images because of deoxyhemoglobin in red blood cells trapped in the thrombus [128]. Two-dimensional TOF techniques are used to evaluate the intracranial venous system because of their excellent sensitivity to slow flow and their diminished sensitivity to signal loss from saturation effects compared with the sensitivities of three-dimensional TOF techniques [131]. CT venography provides a highly detailed depiction of the cerebral venous system and has at least equivalent accuracy for the detection of cerebral venous thrombosis [130].

CT Venography can provide a rapid and reliable modality for detecting IJVthr (Figure 3). CTV is much more useful in subacute or chronic situations because of the varied density in thrombosed sinus.

Magnetic Resonance Venography with time-of-flight (TOF) sequences can also be used as non- invasive diagnostic tool.

Invasive cerebral angiographic procedures are reserved for situations in which the MRI or CT scan results are inconclusive or if an endovascular procedure is being considered. Arteriography findings include the failure of vein appearance due to the occlusion, venous congestion with the enlargement of typically diminutive veins from collateral drainage and reversal of venous flow. The venous phase of cerebral angiography will show a filling defect in the thrombosed cerebral vein/sinus and IJV.

### 3.12. Treatment 

Anticoagulant prophylaxis to prevent IJVthr in patients with central lines has been the subject of multiple clinical trials. Guidelines in cancer patients found six randomized studies investigating the efficacy and safety of vitamin K antagonists vs. placebo or no treatment, one on the efficacy and safety of unfractionated heparin, six on the value of LMWH, one double-blind randomized, and one non-randomized study on thrombolytic drugs and six meta-analyses of thromboprophylaxis in cancer patients with central lines. Considering these data, the use of anti-coagulants for routine prophylaxis of catheter related IJVthr is not recommended [1A]; a central vein catheter should be inserted on the right side, in the jugular vein, and distal extremity of the catheter should be located at the junction of the superior vena cava and the right atrium [132].

The conservative treatment of IJVthr not complicated by CVT is not different from the anticoagulation regimen and duration recommended for deep vein thrombosis in the lower limbs and PE. Management of catheter-related venous thrombosis varies and is not well established, yet [84]. If the catheter can be removed, and there is a low risk of embolization and no prothrombotic risk, the catheter should be removed, and there is no need for anticoagulation therapy. If there is a high risk of embolization, anticoagulant therapy should be initiated [133]. However, current guidelines recommend initial anticoagulation for thrombosis involving proximal upper extremity deep veins (e.g., axillary, subclavian, etc) rather than catheter removal unless anticoagulation is contraindicated [134].

In patients with a major contraindication for anticoagulation (such as recent major hemorrhage), the clinician must balance the risks and benefits of anticoagulation, depending on the clinical situation.

Many invasive therapeutic procedures have been reported such as direct catheter chemical thrombolysis and direct mechanical thrombectomy with or without thrombolysis. However, no randomized controlled trials exist to support these interventions compared with anticoagulation. Catheter directed thrombolysis with or without mechanical thrombo-aspiration can be considered in carefully selected patients. The 2016 ACCP guidelines recommend anticoagulant therapy alone over thrombolysis in patients with acute thrombosis involving the axillary vein and/or the IJV (grade 2C). Thrombolysis can be considered only in patients who meet all the following criteria: severe symptoms, long extent of thrombus, symptoms <14 days, good performance status, life expectancy ≥1 year, and low risk for bleeding [135].

However, most evidence is based on small case series or case reports. From this point of view, a technique for retrograde endovascular recanalization of chronic obstruction from IJVthr has been described [68].

CVT is an uncommon but potentially serious cause of stroke. The anticoagulation therapy is to prevent thrombus growth, facilitate recanalization, and prevent DVT or PE. Risk of cerebral infarction with hemorrhagic transformation or ICH must be considered and may complicate treatment. Randomized controlled trials (RCT) are available on intravenous unfractionated heparin [136] and subcutaneous nadroparin [137].

A systematic review and meta-analysis of 22 studies showed a lower risk of major hemorrhage (1.2% versus 2.1%), thrombotic complications (3.6% versus 5.4%), and death (4.5% versus 6.0%) with LMWH [138]. New direct oral anticoagulants (NOACs), which are target specific for factor Xa (rivaroxaban, apixaban, edoxaban) or thrombin (dabigatran), were studied for CVT treatment. In a multicenter observational study on 111 patients, NOACs appear to be safe and may be as effective as warfarin in patients with CVT [139]. A recently published RCT on 120 patients showed safety and efficacy of dabigatran etexilate 150 mg twice daily for a treatment period of 24 weeks [140]. However, warfarin at a regimen capable of maintaining INR between 2.5 and 3 is still a recommended treatment for CVT, including cases with the combination of extra and intracranial veins thrombosis or the association with inherited or acquired coagulation deficits [126]. The anticoagulation treatment needs to be continued for 3–12 months according to underlying etiology of the thrombosis and to the patient’s risk factors [141].

## 4. Conclusions

To the best of our knowledge, this is the first comprehensive systematic review on IJVthr, which clarifies signs and symptoms, etiology, current diagnosis and treatment of such a severe localization of deep vein thrombosis. The majority of the studies were case reports and/or prospective observational studies, with randomized control trials limited to central venous catheterization and anticoagulation drugs. The comparison of the prevalence of IJVthr in postmortem studies with that reported in clinical articles seems to suggest that IJVthr is an underestimated clinical problem and its incidence increased since the introduction of central venous lines. 

More common symptoms of IJVthr are neck pain and headache, whereas, swelling, erythema and the palpable cord sign beneath the sternocleidomastoid muscle, frequently associated with fever, are the most reported clinical signs.

Ultrasound of the neck, even limited to the simple and rapid CUS assessment, is a quick, economic and effective diagnostic tool, with an elevated sensitivity and specificity. CT and MR venography are second level diagnostic tools and can effectively detect the presence of the thrombosis and even identifying its stage (MRV). Promising non-invasive techniques, like plethysmography, are currently in the developing phase, and aim to screen this condition especially in hospitalized patients with central catheters. Medical treatment of IJVthr resembles that of deep vein thrombosis and PE, whereas the treatment of catheter related IJV thrombosis is not well-established. More data are needed in terms of pharmacological and mechanical endovascular treatment. 

## Figures and Tables

**Figure 1 diagnostics-11-00378-f001:**
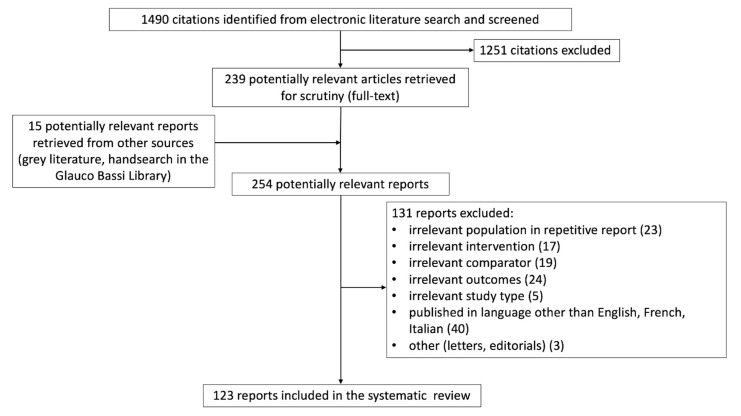
Flowchart of the systematic review, according to PRISMA guidelines [8].

**Figure 2 diagnostics-11-00378-f002:**
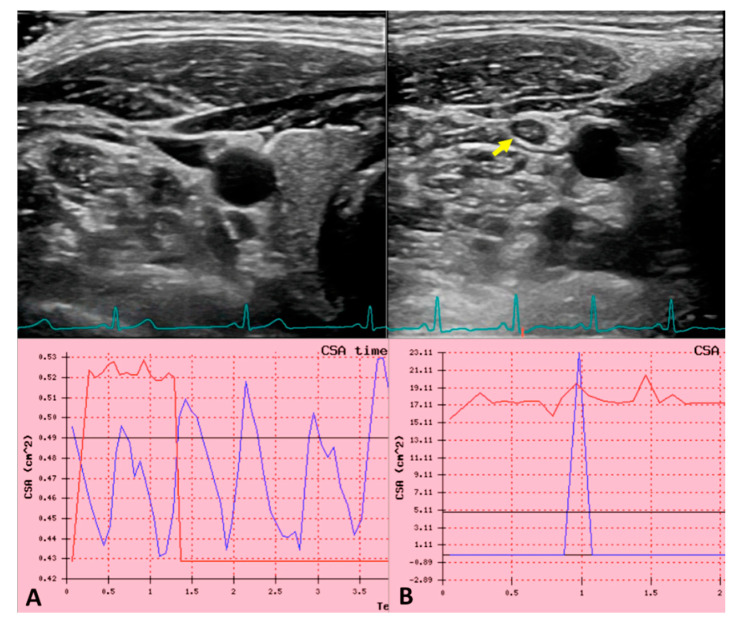
Ultrasound jugular venous pulse in normal and in internal jugular vein thrombosis (IJVthr) cases. (**A**) B-mode ultrasound (US) of the IJV and the common carotid artery at J2 level, in the transversal aspect of the neck. The green line corresponds to the ECG trace. The blue line in the diagram corresponds to the recorded JVP i.e., the sequence of cross-sectional area variation for cardiac beat. The red line corresponds to the synchronized ECG trace. (**B**) The yellow arrow shows an IJVthr. The disappearance of the sequential peaks of the ultrasound JVP is well apparent.

**Figure 3 diagnostics-11-00378-f003:**
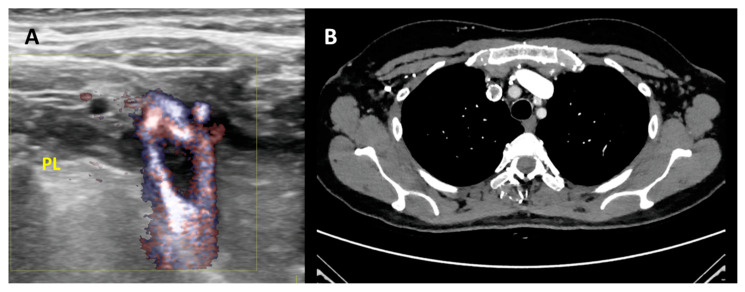
Second level diagnosis of IJVthr. (**A**) Case of right IJVthr with a floating thrombus approaching the superior vena cava; high resolution ultrasound complemented by B-flow enhances the black floating thrombus. PL = pleural line. (**B**) Same case demonstrated by angio-TC.

**Table 1 diagnostics-11-00378-t001:** Signs and symptoms of IJVthr.

Symptoms	Articles
Pain	Gbaguidi [2]; Leci-Tahiri [3]; Carrington [11]; Wada [12]; Al-Zoubi [13]; Gunasekaran [14]; Hahn [15]; Shakeel [28]; Graham [16]
Asymptomatic	Al-Zoubi [13]; Çakir [36]
Difficulty in swallowing, Odinophagia, Disphagia	Carrington [11]; Shakeel [28]
Headache	Duke [17]; Wada [12]; Shakeel [28]; Graham [16]; Thapa [18]; Li C [19]; Ilgen [20]; Hacifazlioglu [21]; Masood [22]
Numbness/somnolence, alteration of consciousness	Duke [17]
Eye discomfort, conjunctival injection	Graham [16]
Blurred vision, decrease visual acuity	Duke [17]; Wada [12]
Dizziness, Nausea	Thapa [18]
Paraesthesia	Wada [12]
**Signs**	
Neck swelling and erythema	Gbaguidi [2]; Leci-Tahiri [3]; Uzun [5]; Bandara [6]; Das [23]; Zhang W [24]; Carrington [11]; Toratani [25]; Wada [12]; Al-Zoubi [13]; Hahn [15]; Shakeel [28]; Bilici [26]; Gianesini [27]; Gunasekaran [14]; Duke [17]
Palpable cord beneath the sternocleidomastoid muscle	Gbaguidi [2]; Leci-Tahiri [3]; Bandara [6]; Shakeel [28]
Superficial varicose collateral veins	Gbaguidi [2]; Mirijello [30]
Indurated vein	Gbaguidi [2]
Fever	Leci-Tahiri [3]; Shakeel [28]; Thapa [18]; Gunasekaran [14]
Oedematous swelling of the face/scalp	Duke [17]; Hyder [29]; Bilici [26]; Thapa [18]; Mirijello [30]
Brachial plexopathy	Glemarec [37]
Horner Syndrome	Glemarec [37]
Papilledema	Duke [17]; Wada [12]; Gunasekaran [14]; Masood [22]
Retinal haemorrhage	Duke [17]
Intracranial hypertension	Duke [17]; Wada [12]; Shakeel [28]; Li C [19]; Masood [22]
VI nerve palsy	Duke [17]
Seizures	Wada [12]
Agraphia	Wada [12]
Progressive Dyspnoea	Bilici [26]
**Laboratory parameters**	
Leucocytosis	Leci-Tahiri [3];Uzun [5]; Toratani [25]; Shakeel [28]; Bilici [26]; Gunasekaran [14]
D-Dimer raising	Toratani [25]; Shakeel [28]; Li C [32]

**Table 2 diagnostics-11-00378-t002:** IJVthr etiology.

Etiology	Articles
Infection	Leci-Tahiri [3]; Hindi [4]; Uzun [5]; Bandara [6]; Li M [32]; Boga [46]; Das [23]; Lin [57]; Duke [17]; Mnejja [62]; Hahn [15]; Shakeel [28]; Graham [16]; Masood [22]
Cervical Neck Fasciitis	Lin [57]
Surgery	Gbaguidi [2]; Leci-Tahiri [3]; Uzun [5]; Li M [32]; Das [23]; Zhang W [24]; Zamboni [49]; Quraishi [50]; Duke [17]; Gunasekaran [14]; Shakeel [28]; Petrov [66]; Mandato [67]; Lupattelli [68]; Graham [16]
Trauma	Gbaguidi [2]; Leci-Tahiri [3]; Hindi [4]; Uzun [5]; Bandara [6]; Zhang W [24]; Duke [17]; Lin D [57]; Mnejja [62]; Gunasekaran [14]; Shakeel [28]
Malignancy	Gbaguidi [2]; Leci-Tahiri [3]; Hindi [4]; Uzun [5]; Bandara [6]; Li M [32]; Das [23]; Zhang W [24]; Duke [17]; Carrington [11]; Toratani [25]; Lin D [57]; Mnejja [62]; Gunasekaran [14]; De Cicco [65]; Harter [63]; Hahn [15]; Shakeel [28]; Siu [69]; Graham [16]; Li [70]; Masood [22]
CVC	Gbaguidi [2] Leci-Tahiri [3]; Hindi [4]; Uzun [5]; Bandara [6]; Giordano [43]; Wilkin [44]; Das [23]; Zhang W [24]; Duke [17]; Toratani [25]; Gunasekaran [14]; De Cicco [65]; Ge [71]; Hyder [29]; Biffi [72]; Biffi [73]; Harter [74]; Hahn [15]; Shakeel [28]; Gheith [75]; Siu [69]; Graham [16]; Li H [70]; Thapa [18]; Mirijello [30]; Masood [22]
Polycythaemia	Leci-Tahiri [3]; Li M [32]; Das [23], Glemarec [37]
Intravenous drug abuse	Leci-Tahiri [3]; Hindi [4]; Bandara [6]; Das [23]; Zhang W [24]; Duke [17]; Merhar [56]; Lin D [57]; Gunasekaran [14]; Shakeel [28]; Graham [16]; Masood [22]
Neck massage/Cervical traction	Leci-Tahiri [3]; Li M [32]; Das [23]; Wada [12]
Ovarian hyperstimulation syndrome	Gbaguidi [2]; Leci-Tahiri [3]; Hindi [4]; Uzun [5]; Das [23]; Lee SH [61], Hahn [15]; Shakeel [28]; Gunasekaran [14]
Hyper-omocysteineimia	Li M [32]; Das [23]; Li H [70]
Hypercoagulable state/thrombophilia/coagulation abnormalities	Gbaguidi [2]; Leci-Tahiri [3]; Hindi [4]; Uzun [5]; Bandara [6]; Das [23]; Bostanci [48]; Zhang W [24]; Duke [17]; Al-Zoubi [13]; Mnejja [62]; Çakir [36]; Gunasekaran [14]; Lim [64]; De Cicco [65]; Shakeel [28]; Mirijello [30]; Ilgen [20]; Masood [22]
Oral contraceptive, hormonal replacement therapy, pregnancy	Gbaguidi [2]; Hahn [15], Masood [22]
Congestive heart failure	Gbaguidi [2]; Uzun [5]
Pacemaker	Gbaguidi [2]; Hindi [4]; Graham [16]
Immobilization	Gbaguidi [2]
Primary (thoracic outlet syndrome, effort, idiopathic)	Gbaguidi [2,49]; Leci-Tahiri [3]; Hindi [4]; Bandara [6]; Toratani [32]; Hahn [15]; Shakeel [28]
Substernal Goitre	Shakeel [28]
Dehydration status and lack of gravitational gradient	Duke [17]; Marshall-Goebel [76]; Zamboni [77]
After balloon venoplasty and/or stenting of the IJV	Mandato [67]; Petrov [66]
Beçhet disease	Hahn [15], Bilici [26]
Truncular IJV malformation	Lim [64]; Lee [61]; Gianesini [27]

## Data Availability

The data presented in this study are available in the article.

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
