# Peer review of "Internal Jugular Vein Thrombosis: Etiology, Symptomatology, Diagnosis and Current Treatment"

_diagnostics, 2021, doi:10.3390/diagnostics11020378_

Round 1

Reviewer 1 Report

Abstract section

Conclusion :

Suggest that write a short conclusion or the outcome of the systemic review.

I have explained in detail in the conclusion section of the manuscript.

Method section:

  • Authors have represented a systemic method for the selection of the studies , and appropriately reviewing and including the final 123 report for the data extraction to write the report.

But there are certain major recommendation that I would suggest.

2.4. Data extraction and quality assessment

 Authors represent that they have assessed the 123 report and they have assessed the 'quality' of the report. Authors state that they have used methodological quality assessment criteria as per the guidelines. The citation of the guidelines that they have mentioned in citation 10 is a case series. (Internal jugular vein stenosis associated with elongated styloid process: five case reports and literature review).

I suggest authors to give the parameters that they have used to assess the studies. 

  • Then from here moving to the "3.2. Study quality" section has not results. There is statement stating included studies was (were) from moderate to good. Recommend that  this section would need further description.

Suggest that make a take suggesting number of studies moderate or good according to there criteria and provide number of the studies used to describe the different aspect like sign symptoms , diagnosis or management.

3.3. Clinical presentation

  • Please explain this last line further "Finally, Table 1 shows that IJV proximal obstruction is a risk factor for non-aneurysm peri-mesencephalic hemorrhage and stroke"

To my understanding the table 1 is just citing the  symptoms and signs.

  • Table 1 includes the serological parameters in the sign , symptoms . Suggest that specify separately i) symptoms ii) signs iii) serological or examination parameters
  • Based on the review of the studies , if authors can comment which is most common sign and which is the most specific presentation than that will be useful extract from the systemic review.

3.4. Post-mortem studies

  • Authors have described one study in this section , citation 17, (Venous Thrombosis and Pulmonary Embolism: A Study of 5039 Autopsies) please mention if there were any other studies on the autopsy analysis.
  • Authors mention that "Pulmonary dissection revealed embolism in 1500 cases (29.8%)" the article has the added information about source of the thrombosis for the PE, like in 12.6% the thrombosis was present in the upper system , this information is useful for the readers. However , it is important to t the DVT of lower extremity is 5 times (59.4%) more common than the upper vein thrombosis.
  • From the article it is very difficult to differentiate if the upper vein thombosis includes the other etiology besides the IJV thr.

3.4. Etiology

Suggest putting the further information regarding the study.

Finally, two systematic reviews assessing femoral versus IJV routes didn’t find significant differences in catheter coloni-146 zation, catheter-related bloodstream infection and thrombotic complications [53,54].

Please represent if there is difference in the rate of the thrombosis

"However, findings of the same study 152 recommend to prefer the IJV for female, non-chlorhexidine-impregnated dressings users, 153 and when catheters are left in place more than 4 days [55]."

Is there any literature to suggest that the catheter related thrombosis is more common than the non-catheter related etiology

3.6. Pulmonary embolism and other serious complications

      • Please provide a citation for this data

The mean rate of PE occurring in case of IJVthr is 5%.

      • Please present the  rate of the PE in the patient with the catheter related thrombosis

3.10. Second level diagnosis

Please provide the citation and explained the difference in the sensitivity of the CTV vs MRI  for the following paragraph

"MRI is more sensitive than CT at each stage after thrombosis. The MR signal intensity 308 of venous thrombus varies according to the time of imaging from the onset of thrombus  formation."

Please given your comment on whether a patient with the unvealing IJV thrombus should have the MRI done for the further evaluation.

3.11. Treatment

Treatment differs for the patient with the catheter related thrombosis vs those with no catheter.

In a patient with the IJV thrombosis and removal of catheter does not need further intervention. This is should be explained with citation.

  1. Conclusions
    • Please present with important conclusion drawn from your study like
    1. Post-mortum rate
    2. Common signs and symptoms
    3. Sensitivity of CUS, CTV
    4. Best imaging studies modality
    5. Treatment  in patient with catheter related IJV thrombosis

Reviewer 2 Report

The Study is of excellent level, together very original and unusual, absolutely to be kept in mind. The issue is incredibly underestimated but, so far, nobody took the time to estimate how common it can be. I think it is also a good opportunity to open a door towards safe diagnostic exams that I freely recommend as "routine exams" within certain wards and type of patients.

The suggestion is to introduce this issue and diagnostic methods as much as possible. 

Reviewer 3 Report

The authors present a detailed meta-analysis of a topic of high interest in clinical practice. Authors have done an excellent job purging the articles with information that is not useful for the purpose of the article and present an article that is well written, well thought out and very useful to readers.

Round 2

Reviewer 1 Report

Authors have incorporated all the suggestion and significantly improved upon the manuscripts. I suggest accepting the manuscript. I thank the authors for the significant contribution

Author Response

We thank this reviewer for the suggestion which, actually,  improves our manuscript.